# Platelet Endothelial Cell Adhesion Molecule 1 (CD31) Is Essential for *Clostridium perfringens* Beta-Toxin Mediated Cytotoxicity in Human Endothelial and Monocytic Cells

**DOI:** 10.3390/toxins13120893

**Published:** 2021-12-13

**Authors:** Basma Tarek, Julia Bruggisser, Filippo Cattalani, Horst Posthaus

**Affiliations:** 1Institute of Animal Pathology, Vetsuisse Faculty, University of Bern, 3012 Bern, Switzerland; basma.tarek@vetsuisse.unibe.ch (B.T.); julia.bruggisser@icloud.com (J.B.); filippo.cattalanitognola@vetsuisse.unibe.ch (F.C.); 2Department of Pathology, Faculty of Veterinary Medicine, Aswan University, Aswan 81528, Egypt

**Keywords:** PECAM-1, CD31, *Clostridium perfringens*, beta-toxin, pore forming toxin, membrane receptor, endothelial cell, monocyte, human, CRISPR/Cas9

## Abstract

Beta toxin (CPB) is a small hemolysin beta pore-forming toxin (β-PFT) produced by *Clostridium perfringens* type C. It plays a central role in the pathogenesis of necro-hemorrhagic enteritis in young animals and humans via targeting intestinal endothelial cells. We recently identified the membrane protein CD31 (PECAM-1) as the receptor for CPB on mouse endothelial cells. We now assess the role of CD31 in CPB cytotoxicity against human endothelial and monocytic cells using a CRISPR/Cas9 gene knockout and an antibody blocking approach. CD31 knockout human endothelial and monocytic cells were resistant to CPB and CPB oligomers only formed in CD31-expressing cells. CD31 knockout endothelial and monocytic cells could be selectively enriched out of a polyclonal cell population by exposing them to CPB. Moreover, antibody mediated blocking of the extracellular Ig6 domain of CD31 abolished CPB cytotoxicity and oligomer formation in endothelial and monocytic cells. In conclusion, this study confirms the role of CD31 as a receptor of CPB on human endothelial and monocytic cells. Specific interaction with the CD31 molecule can thus explain the cell type specificity of CPB observed in vitro and corresponds to in vivo observations in naturally diseased animals.

## 1. Introduction

*Clostridium perfringens* beta-toxin (CPB) is a highly potent exotoxin produced by *Clostridium perfringens* type C. It is essential for the induction of a fatal acute necro-hemorrhagic enteritis (NE) in newborn animals, particularly in piglets [1,2]. In humans, type C enteritis was a significant cause of childhood lethality in the highlands of Papua New Guinea until a successful vaccination program was implemented [3]. Nowadays, reports in humans are sporadic [4,5,6,7].

CPB is produced as a monomeric protein with a molecular weight of 34.86 kDa [8]. It belongs to the small beta–pore-forming toxins (β-PFT) of the hemolysin family [9]. β-PFTs are secreted as water-soluble monomers that bind to membrane receptors on their target cells. Surface binding leads to an increase in the local concentration of β-PFTs, oligomerization and the assembly of a prepore complex that subsequently inserts into the lipid bilayer and forms a functional transmembrane pore [10,11,12]. The pores disturb membrane permeability by allowing the passage of ions leading to intracellular concentration changes and responses of the affected cells.

CPB damages the endothelium of mucosal vessels in the small intestine [13,14,15]. In vitro, the toxicity of CPB is remarkably specific toward endothelial cells, platelets and different human cell lines of the hematopoietic lineage, such as the monocytic and myelocytic THP-1 and U937 cells [15,16,17,18,19]. We recently identified Platelet Endothelial Cell Adhesion Molecule-1 (CD31 or PECAM-1) as a specific membrane receptor for CPB on mouse endothelial cells [19]. In addition, we discovered that the extracellular membrane proximal Ig6 domain of CD31 is essential for the interaction with CPB. Conservation of this domain between susceptible species, such as mice, humans and pigs, predicts the targeting of CD31 by CPB in different host species. Because CPB targets other CD31-expressing cells [19], we additionally wanted to prove the essential function of CD31 in human monocytic cells lines. In the present study, we demonstrate that CD31 functions as the membrane receptor for CPB on human endothelial and monocytic cell lines using a CRISPR/Cas9 gene knockout approach and antibody mediated blocking of the Ig6 domain of human CD31.

## 2. Results and Discussion

### 2.1. Human Endothelial and Monocytic Cell Lines Express CD31 and Are Susceptible to CPB

For our study, we used the human endothelial cell line HMEC-1, the human endothelial somatic cell hybrid line EA.hy926, the human monocytic leukemia cell line THP-1 and the human monocytic-like cell line U937 (derived from a histiocytic lymphoma). Cell viability tests confirmed previous results from our and other groups that these cells are sensitive to the cytotoxic effects of CPB (Figure 1A,C) [17,19,20,21]. We verified CD31 protein expression by Western blotting (Figure 1B,D). 

### 2.2. CD31 Is Essential for CPB-Mediated Cytotoxicity in Human Endothelial and Monocytic Cells

To investigate whether CD31 expression is essential for CPB toxicity in human endothelial and monocytic cell lines, we employed a CRISPR/Cas9 single gene knockout approach. First, we introduced Cas9 and two independent guide RNAs (gRNAs) targeting exon 5 and 6 of the CD31 gene named gRNA1 (gR.1) and gRNA2 (gR.2), respectively, into HMEC-1 cells. After puromycin selection, the resulting polyclonal cell populations were sequenced and the editing efficiency of each gRNA were compared with cells transduced with non-targeting gRNA (gR.NT) using TIDE analysis. This showed that the gRNA targeting exon 6 (gR.2) was efficient in generating frameshift mutations in the CD31 gene (Figure 2A). Next, we transduced all other cell lines using this gRNA. This resulted in polyclonal cell populations with 79–92% frameshift mutations in the CD31 gene (Figure 2B) and a substantial decrease in CD31 protein expression levels (Figure 2C). Cell viability tests with these polyclonal cell populations confirmed that the loss of CD31 expression resulted in significantly increased resistance to CPB (Figure 2D). Western blotting showed that CPB oligomer formation depends on the expression of CD31 (Figure 2E). In combination with the cell viability tests, this clearly indicates that cell damage by CPB depends on CD31 expression in the investigated cell lines.

We rescued EA.hy926∆CD31gR.2 cells with full-length human CD31-GFP (EA.hy926∆CD31gR.2/CD31), modified by four silent mutations to avoid further editing by Cas9. CD31-GFP expression restored the susceptibility to CPB (Figure 3A,B). We were however unable to restore CD31-GFP expression in HMEC-1 and both monocytic cell lines because these cells stopped proliferating after transductions. We therefore ectopically expressed the same construct in HEK293FT cells, which lack endogenous CD31. This rendered these cells susceptible to CPB (Figure 3C,D). In addition, oligomer formation of CPB only occurred in HEK293huCD31-GFP cells (Figure 3E). 

### 2.3. Selective Enrichment of CD31 Knockout Cells with CPB

To further prove the importance of CD31 in susceptibility to CPB, we tested whether we could select HMEC-1 cells that lost CD31 expression out of a polyclonal cell population. To this purpose, we incubated HMEC-1 cells targeted with the less efficient gR.1 (9% frameshift mutations) with a cytotoxic dose of CPB (1 µg/mL for 24 h). After the first exposure of the CD31gR.1 cells to CPB, a moderate number of surviving cells were observed after 24 h (Figure 4A). These cells were then grown to confluence, sequenced and TIDE analysis was performed. Results showed a marked enrichment of cells carrying frameshift mutations in the CD31 gene from 9% to 79.2% (Figure 4B). These cells were then largely resistant to a second incubation with the toxin (Figure 4A). This was accompanied by a loss of CD31 expression in HMEC-1ΔCD31gR.1 cells under CPB selection (Figure 4C). We next repeated the same experiment with THP-1 cells, obtaining very similar results (Figure 4D–F).

Taken together, our findings highlight the essential role of CD31 in CPB-mediated cytotoxicity. Using a CRISPR/Cas9 knockout approach, we could confirm our previous results obtained with CD31 knockout mouse endothelioma cells in human endothelial cells. Importantly, we could also show the essential role of CD31 for CPB cytotoxicity in human monocytic cell lines. CD31 is a 130 kDa type 1 transmembrane glycoprotein that belongs to the Immunoglobulin (Ig)-superfamily of cell adhesion molecules [22]. It is highly expressed in endothelial cells, where it is a constituent of cell junctions and concentrated at the lateral cell borders [23,24]. In addition, CD31 is also expressed on the surface of non-erythroid cells of the hematopoietic lineage (platelets, macrophages, monocytes, neutrophils, T- and B-Lymphocytes) [23]. In contrast, CD31 is absent from epithelial cells, fibroblast and red blood cells [25]. Thus, cellular expression of CD31 overlaps with the in vitro cell type specificity of CPB reported so far [11,17,18,19,21,26]. This observation correlates with in vivo findings, where CPB has been shown to target vascular endothelial cells in the small intestine of affected hosts [13,14]. Although epithelial damage has been shown to occur rapidly in infection models of *C. perfringens* type C enteritis [27], currently there is no clear evidence that CPB directly damages epithelial cells through pore-formation [28]. As our study concentrated on the role of CD31, we cannot exclude that CPB can bind to alternative receptors on other cells. Nagahama et al. reported that the widely expressed membrane molecule P2X7 could function as a CPB receptor on monocytes and other cells [29]. We therefore investigated the P2X7 expression of all cells used in our study by Western blotting and found no alterations in P2X7 protein levels that could account for the different susceptibilities of CD31 proficient and deficient cells (Appendix A). More detailed studies to determine the potential interaction between CPB and P2X7 would thus be necessary to explain this discrepancy between our results and those reported by Nagahama et al. [29].

### 2.4. Blocking the CD31 Ig6 Domain Protects Sensitive Cells against CPB Induced Cytotoxicity

CD31 is composed of a large extracellular domain containing six Ig-like domains, followed by a single pass α-helical transmembrane domain and a cytoplasmic tail of variable length [30]. The cytoplasmic domain participates in signaling functions. CD31′s adhesive properties are mediated by its extracellular domain via binding to other CD31 ectodomains (homophilic adhesion) or other ligands (heterophilic adhesion) [24]. CD31/CD31 homophilic interactions are mediated by the Ig1 and Ig2 domain [31,32,33,34].

They include either cis interactions between adjacent CD31 molecules in the same membrane or trans interactions between individual CD31 from adjacent cells [34,35]. The latter are essential for concentrating CD31 at the endothelial cell borders. Here, CD31 contributes to cell junctional integrity and mediates neutrophil diapedesis across the vascular wall [36]. CD31 also binds to other ligands, including integrin α_V_β_3_ (present on platelets), CD38 and CD177 (present on leukocytes) via its Ig6 domain [37,38,39]. The binding interaction between CD177 on neutrophils and CD31 on endothelial cells has been shown to be important for CD31-mediated leukocyte transmigration [40]. We had previously determined that the extracellular Ig6 domain of mouse CD31 is essential for the interaction of the receptor with CPB, and hypothesized that this domain contains the binding site for CPB. Chimeric proteins of mouse CD31 containing the human and porcine CD31 Ig6 domain also interacted with CPB, indicating conservation of the receptor binding region in these three species [19]. To further consolidate that human CD31 serves as a membrane receptor for CPB via its Ig6 domain, we now employed a previously described monoclonal antibody (mAb) blocking approach using three distinct human CD31 mAbs: one blocking the Ig1 domain (mAb 1.3) and two blocking the Ig6 domain (mAbs 1.2 and 4G6) [33]. EA.hy926 and THP-1 cells pre-incubated with antibodies blocking the CD31 Ig6 domain became resistant to CPB, whereas blocking the CD31 Ig1 domain had no effect on CPB cytotoxicity (Figure 5A,C,D). Moreover, Ig6 domain blocking abolished the formation of SDS-resistant oligomers in these cells, demonstrating that pore formation in the plasma membrane was prevented (Figure 5B,E). The epitope for the mAb4G6 has been localized to a hexapeptide sequence (CAVNEG) located at AA position 523–528 [41] (signal peptide included in numbering). Because this sequence is not conserved between humans and mice, this antibody only blocks human Ig6 function. Importantly, the epitope is located directly proximal to the region that we previously identified to be essential for CPB interaction across mouse, human and porcine CD31 Ig6 using mutational studies (Appendix A) [19]. Yan et al. [41] showed that the 4G6 epitope is at least partially conformation dependent as antibody binding to this region was influenced by amino acid exchanges in a 15 amino acid region (CAVNEGSGPITYKFY). The last two amino acids of this region overlap with the one shown by us to be essential for interacting with CPB [19]. Ig domains consist of seven to nine antiparallel β-strands arranged into two sheets linked by a disulfide bond [42]. Assuming a similar fold of Ig6 as in the Ig1 and Ig2 domains [31], the entire region now defined by two independent approaches seems to be important for CPB cytotoxicity and oligomer formation. This region is located in a surface-exposed loop between two beta strands of the Ig 6 domain. It is thus accessible for potential binding partners. Binding of the antibody to this region could inhibit CD31-CPB interaction by directly blocking the binding site, induction of conformational changes affecting the binding site, or by steric hindrance. So far, neither the structure of the CD31 Ig6 domain nor that of CPB have been clearly defined. In addition, we currently do not know whether the monomer and/or the oligomeric form of CPB bind to the CD31 molecule. Although we were able to co-immunoprecipitate CPB oligomers with CD31 in our previous study [19], interaction with the monomer and subsequent oligomerization could be so rapid that we are unable to detect interaction with monomeric CPB using this method. Determining the structure of CPB and the CD31 Ig6 domain will be essential to investigate the molecular basis of the CPB–receptor interaction.

## 3. Conclusions

This study confirms the essential role of CD31 (PECAM-1) for CPB-mediated cytotoxicity in human endothelial and monocytic cells using a CRISPR/Cas9 gene knockout approach. Using antibody blocking of CD31 extracellular Ig domains, we further substantiate the notion that a particular surface exposed region in the Ig6 membrane proximal domain plays an essential role in the interaction with CPB. This confirms our previous results where we could unambiguously show that CD31 is the cell type specific receptor for CPB on mouse endothelial cells. Extending these results to CD31-expressing monocytic cells supports the hypothesis, that the specific interaction of the toxin with this membrane receptor accounts for the remarkable cell type specificity of CPB observed in vitro. The specific expression of CD31 on endothelial and hematopoietic lineage cells make this molecule a plausible target for *C. perfringens* type C. Besides vascular damage and hemorrhage that produces favorable growth conditions for this bacterium, targeting phagocytes and potentially other CD31 expressing immune cells could help the pathogen to evade the immune response of the host.

## 4. Materials and Methods

### 4.1. Cells and Reagents

HMEC-1 (ATCC, CRL-3243™) and EA.hy926 (ATCC, CRL-2922™) were cultivated in MCDB 131 Medium (Gibco, product 10372019) supplemented with 10% fetal bovine serum FBS (Gibco), 10 ng/mL epidermal growth factor EGF (Gibco), 10 mM L-Glutamine (Gibco), 1 μg/mL Hydrocortisone (Sigma Aldrich, St. Louis, MO, USA), in the presence of 1% penicillin-streptomycin (Gibco) grown at 37 °C in an atmosphere containing 5% CO_2_. HEK293FT cells were cultured in DMEM medium (Gibco, product 41965-039) supplemented with 10% FBS and 1% penicillin-streptomycin. U937 and THP-1 cells (kindly provided by Dr. René Köffel, Institute of Anatomy, University of Bern, Switzerland) were grown in RPMI 1640 medium (Gibco, product 11875-093) supplemented with 10% FBS and 1% penicillin-streptomycin.

### 4.2. Recombinant CPB Production

For cell culture and in vitro experiments, recombinant CPB was designed as N-terminally His-Tagged protein (Appendix A) and the codon optimized gene was ordered from GenScript^®^ (Piscataway, NJ, USA). Recombinant CPB toxin was produced and purified by the Protein Production and Structure Core Facility of the EPFL (Lausanne, Switzerland). CPB concentrations were determined using the Pierce™ BCA protein assay Kit (ThermoFischer Scientific, Waltham, MA, USA) and confirmed by SDS-PAGE. Activity of each batch was tested prior experiments on primary porcine endothelial cell cultures as described [43].

### 4.3. Western Blotting

Cells were washed with PBS and lysed with modified RIPA buffer (25 mM Tris-HCl pH 7.4, 150 mM NaCl, 1% NP-40, 1% Na-Deoxycholate, 0.1% SDS and complete protease inhibitor cocktail EDTA-free (Roche, Basel, Switzerland)) for 60 min on ice. Lysates were cleared using centrifugation (16,000 RCF, 20 min 4 °C). Total protein concentration was determined using the Pierce^TM^ BCA protein assay kit (ThermoFischer Scientific) and the supernatants were adjusted to 2% SDS in reducing gel sample buffer and boiled for 5 min. Protein samples (40 μg) were separated by 7–10% acrylamide Tris-glycine SDS-PAGE and transferred onto 0.45 μm nitrocellulose membranes (GE Healthcare). Membranes were blocked in blocking buffer (Intercept^®^ PBS Blocking Buffer, Li-COR) and incubated overnight with primary antibodies at 4 °C. The monoclonal antibodies used were mouse anti-CD31 (Santa Cruz, H-3, dilution 1:500), mouse anti-CPB (10A2, USDA, 1:1000) and mouse anti-α-tubulin (Sigma-Aldrich clone DM1A, dilution 1:5000). The polyclonal antibody used was rabbit anti-P2X7 (Almone labs APR-004, 1:300). Antibodies were detected with an Azure imaging system (c600) using IRDye secondary antibodies (Li-COR Biosciences, dilution 1:5000).

### 4.4. SDS-Resistant Oligomer Formation

Cells were incubated with 8 µg/mL CPB for 20 min at 37 °C for adherent cells and 10 µg/mL CPB for 60 min at 37 °C for suspended cells. Cells were lysed in modified RIPA buffer as described above, supernatants were adjusted to 2% SDS in reducing gel sample buffer and boiled for 5 min. Protein samples (100 µg) were separated by 7% acrylamide Tris-glycine SDS-PAGE and analyzed by Western blotting.

### 4.5. Cell Viability Assays

Effects of CPB on cells were measured using CellTiter-Blue^®^ (Promega Corporation, Madison, WI, USA) Cell Viability Assay (Promega #G8080). Cells (30,000 cells/cm^2^) were grown to confluency in a 96-well plate and incubated with CPB for 24 h. Resazurin dye was added to a 0.002% final concentration, incubated for 2 h at 37 °C and fluorescent signal intensity was quantified using the EnSpire Multimode Plate Reader (PerkinElmer, Waltham, MA, USA) at excitation and emission wavelengths of 540 and 612 nm, respectively. Viability of cell lines was calculated as percentage of average values determined with untreated control cells (*n* = 4) according to the following formula: (Florescence value of the well)/(average value of 4 control cell wells) × 100.

For visualization of cytopathic effects of adherent endothelial cell lines, cells were seeded in 96-well plates, grown to confluence and exposed to CPB (1 µg/mL) for 24 h, fixed with methanol and stained with Eosin/Azur dye (Hemacolor^®^ Rapid staining of blood smear, Merck KGaA, Darmstadt, Germany). Cytopathic effects were evaluated by light microscopy with 200× magnification. For THP-1 cells, images of unfixed cells were taken by light microscopy with 200× magnification.

### 4.6. Generation of CD31 Knockout Cell Lines

CD31 was knocked out in HMEC-1, EA.hy926, THP-1 and U937 using CRISPR/Cas9 system. We adopted two different gRNA targets for human CD31 gene named CD31 gR.1 (CTTCCATGATCATTCCGGTG) and CD31 gR.2 (ATAGCCTCAAAGTCGGACAG) and a non-target sequence for negative control (TGATTGGGGGTCGTTCGCCA) from the human Brunello library [44]. Forward and reverse single-stranded oligonucleotides of each gRNA were ordered from Microsynth (Balgach, Switzerland) and were annealed to generate a double-stranded oligonucleotide. These target sequences were cloned into the plentiCRISPR v2 (Addgene #52961) which contains sgRNA, Cas9 and a puromycin resistance cassette. The vector was transformed into Endura chemically competent cells (Lucigen #60240). Bacteria were grown on LB agar plates with carbenicillin and incubated at 37 °C overnight. Clones were selected and grown in LB broth overnight. Plasmids were purified and sequenced with the U6 forward primer (GAGGGCCTATTTCCCATGATTCC) to verify the presence and proper orientation of double-stranded oligonucleotide. HEK293FT cells were plated at a density of 57,000 cells/cm^2^ for 24 h prior to transfection in a 150 mm cell culture dish. Transfection was performed using a calcium phosphate precipitation protocol. Briefly, we mixed the 32 µg Transfer vector (lentiCRISPR v2 gRNA) with the following packaging plasmids: 9 μg pCMV-VSVG (Addgene #8454), 12.5 μg pMDLg/pRRE (Addgene #12251) and 6.25 μg RSV-Rev (Plasmid #12253). The plasmid solution made up to a final volume of 1125 μL of 0.1× TE/dH_2_O. 125 μL of 2.5M CaCl_2_ was added, and the mixture was incubated at RT for 5 min. After incubation, we added 2× HBS dropwise to the mix while vortexing at full speed. The transfection mixture was added dropwise to the cells. The growth medium was refreshed 16 h post transfection. The virus-containing media were harvested and cleared by filtration through 0.45 μm cellulose acetate filters (GVS Filter Technology, Sanford, ME, USA) 48 and 72 h post-transfection. Lentivirus containing media were concentrated by ultracentrifugation at 22 k rpm for 2 h, 4 °C. The lentivirus pellets were resuspended in PBS and stored at −80 °C. qPCR Lentivirus Titer Kit (abm # LV900) was used to measure the concentration of the viruses using the ABI^®^ 7500 Fast real time PCR system. For the transduction, cells were seeded at a density of 26,000 cells/cm^2^ in a six-well plate and cultivated for 24 h. The culture medium was replaced by 4 mL of 8 μg/mL polybrene containing medium and along with lentivirus at a MOI of 10. After incubation for 48 h, the growth medium was refreshed. For selection of CRISPR-targeted cells, puromycin (2.5 μg/mL in EA.hy926 and U937 cells, 2 μg/mL in THP-1 cells and 1 μg/mL in HMEC-1 cells) was added 48 h post-transduction and maintained for 3 days. To determine the efficiency of the integration of each gRNA, the TIDE analysis tool was used [45]. We isolated gDNA of each cell lines using DNeasy Blood & Tissue Kit (Qiagen # 69504). gRNA sequences that were integrated into the genomes of the cells were amplified using primers annealing up- and downstream of the gRNA sequence. The PCR products were purified using NucleoSpin Gel and PCR Clean-up (Macherey Nagel # 740609) and sequenced in Microsynth (Balgach, Switzerland).

### 4.7. Complementation and Overexpression of CD31

The sequence of human CD31 gR.2 was modified by Q5^®^ Site-Directed Mutagenesis Kit (NEB #E0554S) in the 20 nucleotide CRISPR region with four silent mutations (highlighted in bold) without changing the amino acid sequence (from ATAGCCTCAAAGTCGGACAG to ATTGCGTCAAAATCGGATAG). The modified sequences were synthesized and cloned into pLenti-P2A-GFP-Blast plasmid (modified from pLenti-P2A-Puro, Origene #PS100109). PLenti-P2A-GFP-Blast with huCD31 with mutated human CD31 gR.2 was used to generate lentivirus as described above and EA.hy926∆CD31gR.2 and HEK293FT cells were transduced. After transduction, cells were selected with 6 μg/mL and 5 μg/mL blasticidin S hydrochloride (Sigma Aldrich #15205) in EA.hy926 and HEK293FT, respectively.

### 4.8. CD31 Extracellular Ig Domains Blocking Assay

For the viability assays, EA.hy926 cells were seeded in 96-well plates at a density of 93,000 cells/cm2 one day before the experiment. The experiment was performed as previously described [33]. Human CD31 monoclonal antibodies were kindly provided by Prof. Peter J. Newman, Medical College of Wisconsin, USA. Briefly, cells in each well were incubated with 100 µL of medium containing 20 µg of one of the following monoclonal antibodies: CD31-1.2 and 4G6 (CD31 Ig domain 6) and CD31-1.3 (CD31 Ig domain 1) [41]. Following 60 min incubation at 37 °C, cells were washed 2 times with PBS and incubated with 1 μg/mL CPB for 24 h at 37 °C. THP-1 cells (300,000/condition) were incubated in a medium containing 20 µg of human CD31 mAbs for 60 min. Cells then were washed twice with PBS and resuspended in medium and were seeded in 96-well plate in a density of 93,000 cells/cm2 on the same day of the experiment. 1 μg/mL CPB for 24 h at 37 °C were added to the cells. Effects of CPB on cells were measured using Resazurin assay as mentioned previously. For visualization of cytopathic effects of EA.hy926, cells were seeded in Nunc^®^ Lab-Tek^®^ chamber slide^TM^ system, in the same conditions as above and were preincubated with 20 µg of human CD31 mAbs for 1 h, 37 °C. Then, cells were exposed to CPB (1 µg/mL) for 24 h, fixed with methanol and stained with Eosin/Azur dye (Hemacolor^®^ Rapid staining of blood smear, Merck KGaA). Cytopathic effects were evaluated by light microscopy with 200× magnification. For SDS-resistant oligomer formation assay, EA.hy926 were seeded in a density of 40,000/cm^2^ in T25 and grown to confluency and 10 million cells of THP-1 cells were used. Cells were incubated with a medium containing 20 µg of mAbs. Following 60 min incubation at 37 °C, SDS-resistant oligomer formation assay and Western blotting was performed as described above. All experiments were conducted in two biological replicates.

### 4.9. Immunofluorescence

Cells grown to confluency on 0.2% gelatin coated glass coverslips were washed with PBS and subsequently resuspended in ice-cold PBS. All further steps were performed in a wet chamber. Cells were fixed for 10 min in 4% paraformaldehyde in PBS, pH 7.2 at room temperature and washed with ice-cold PBS. Cells were incubated in PBS containing 0.2% Triton X-100 for 10 min and blocked for 45 min with 3% BSA at room temperature. Cells were rinsed in ice-cold buffer and DNA was stained with Hoechst (ThermoFischer Scientific) and mounted with Fluorescent Mounting medium (Dako). Images were recorded with a Nikon Eclipse 80i Fluorescent Microscope.

### 4.10. Quantification and Statistical Analysis

Statistics were performed using GraphPad Prism 9. Statistical details of experiments can be found in figure legends. For all multiple comparison two-way ANOVA, Sidak’s multiple comparison test, a 99.9% confidence interval was used.

## Figures and Tables

**Figure 1 toxins-13-00893-f001:**
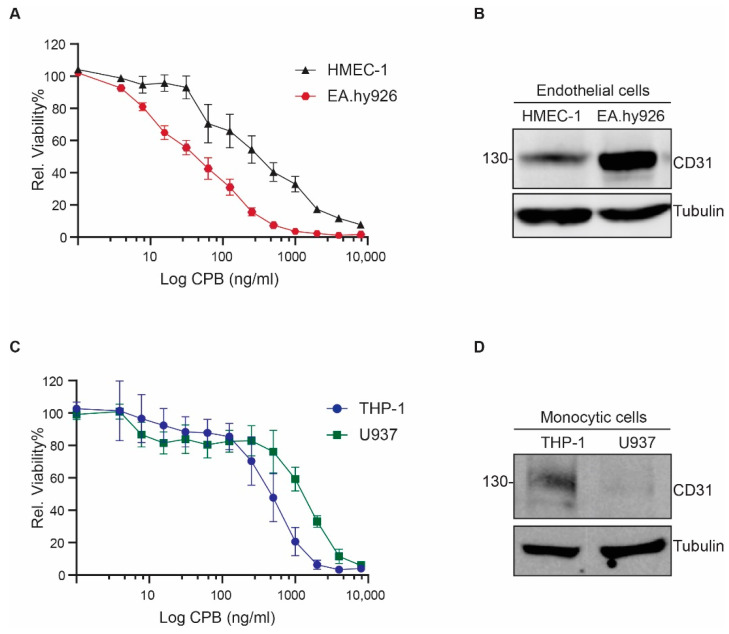
Susceptibility of human endothelial and monocytic cells to CPB. (**A**) Viability of HMEC-1 and EA.hy926 incubated with the indicated concentrations of CPB (24 h, 37 °C) as a percentage of untreated control cells. Data are represented as means (*n* = 8) ± SD. (**B**) Western blots of HMEC-1 and EA.hy926 whole cell lysates showing CD31 protein expression. Tubulin served as a loading control. (**C**) Viability of THP1 and U937 incubated with the indicated concentrations of CPB (24 h, 37 °C) as a percentage of untreated control cells. Data are represented as means (*n* = 8) ± SD. (**D**) Western blots of THP-1 and U937 whole cell lysates showing CD31 protein expression. Tubulin served as a loading control.

**Figure 2 toxins-13-00893-f002:**
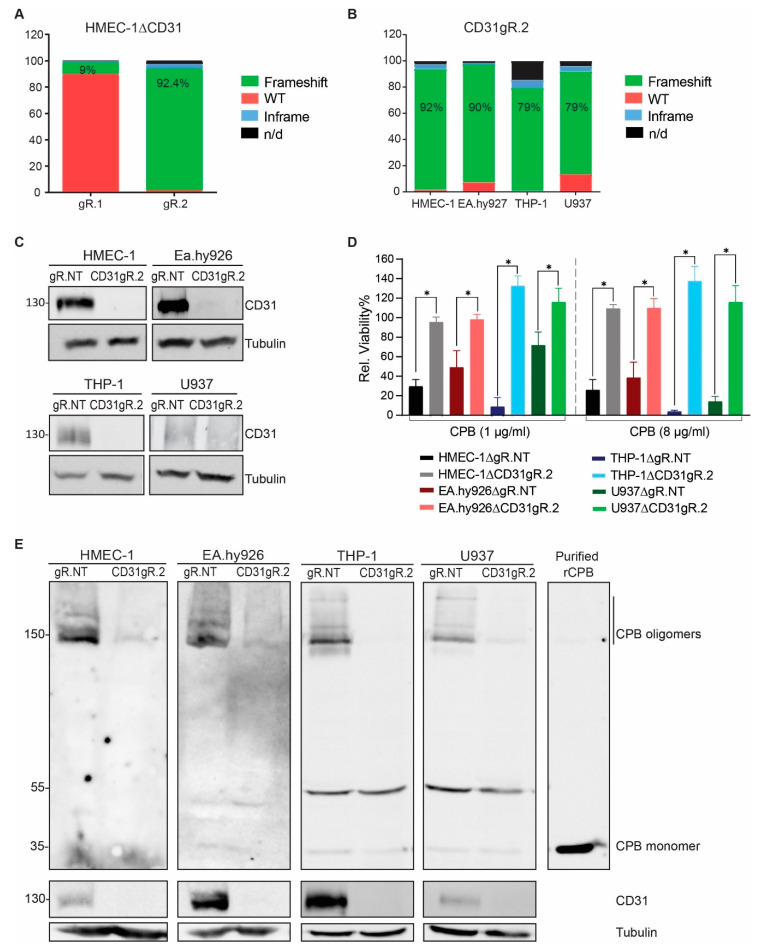
CD31 knockout cells are resistant to CPB. (**A**) TIDE analysis of HMEC-1∆CD31 gR.1 and gR.2 (**B**) TIDE analysis of CD31gR.2 in HMEC-1, EA.hy926, THP-1 and U937. (**C**) Western blots of HMEC-1, EA.hy926, THP-1 and U937∆CD31gR.2 polyclonal cell populations. As control, cells transduced with gR.NT were used. Tubulin served as loading control. (**D**) Viability of CD31gR.2 transduced cells incubated with 1 µg/mL or 8 µg/mL CPB (24 h/37 °C) as a percentage of untreated control cells. Multiple comparison two-way ANOVA, Sidak’s multiple comparison test, values with *p* < 0.0001 are indicated by an asterisk. Data are represented as means (*n* = 8) ± SD. (**E**) Western blots of RIPA lysates of HMEC-1, EA.hy926, THP-1 and U937∆CD31gR.2 polyclonal cell populations incubated with 8 µg/mL CPB (20 min, 37 °C) and purified recombinant CPB probed with anti-CPB and anti-CD31 antibodies. Tubulin served as loading control. 55 kDa band in THP-1 and U937 panels are remnant of tubulin signal.

**Figure 3 toxins-13-00893-f003:**
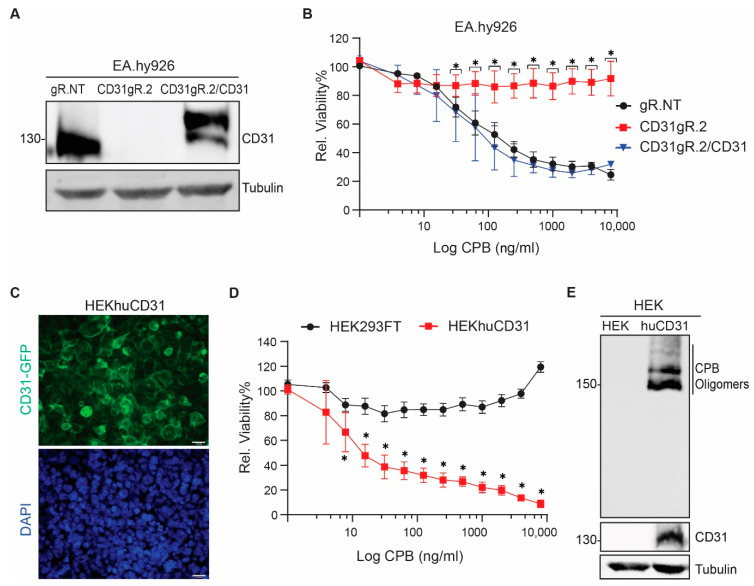
Reconstitution of CD31 expression restores CPB susceptibility. (**A**) Western blots of EA.hy926∆gR.NT, EA.hy926∆CD31gR.2 and EA.hy926∆CD31gR.2 cells re-transduced with CD31-GFP (EA.hy926∆CD31gR.2/CD31) probed with the indicated antibodies. Tubulin served as loading control. (**B**) Viability of EA.hy926∆gR.NT, EA.hy926∆CD31gR.2 and EA.hy926∆CD31gR./CD31 incubated with indicated concentrations of CPB. Viability data are represented as means (*n* = 8) ± SD. Multiple comparison two-way ANOVA, Sidak’s multiple comparison test, values with *p* < 0.0001 are indicated by an asterisk. (**C**) Immunofluorescence of HEK293FT cells transduced huCD31-GFP (HEKhuCD31) indicating membrane localization of ectopically expressed CD31 (400× magnification). (**D**) Viability of non-transduced HEK293FT and HEKhuCD31 incubated with indicated concentration of CPB (24 h, 37 °C). Viability data are represented as means (*n* = 8) ± SD. Multiple comparison two-way ANOVA, Sidak’s multiple comparison test, values with *p* < 0.0001 are indicated by an asterisk. (**E**) Western blots of non-transduced HEK293FT and HEKhuCD31 incubated with 8 μg/mL CPB for 20 min at 37 °C and probed with anti-CPB and anti-CD31 antibodies. Tubulin served as loading control.

**Figure 4 toxins-13-00893-f004:**
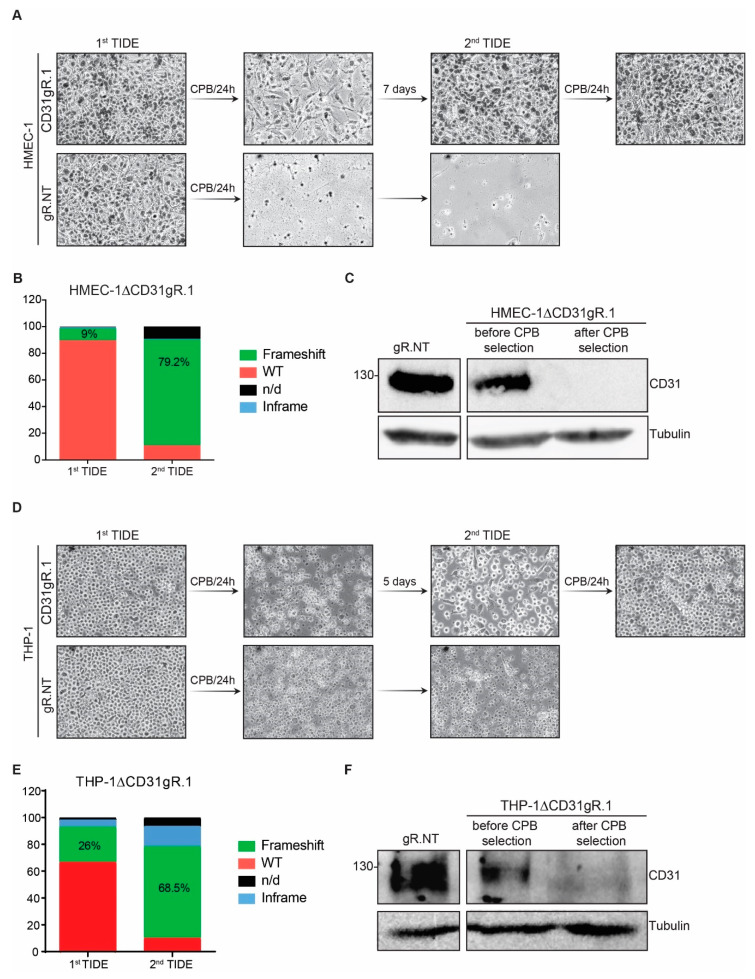
Enrichment of CD31 knockout cells using CPB. (**A**) Representative micrographs of HMEC-1∆CD31gR.1 and HMEC-1gR.NT cells during a selection process using two consecutive incubations with CPB (1 μg/mL, 24 h, 37 °C) (200× magnification). (**B**) TIDE analysis of HMEC-1∆CD31gR.1 before and after selection with CPB. As control, cells transduced with gR.NT were used. (**C**) Western blots of HMEC-1∆gR.NT and HMEC-1∆CD31gR.1 before and after selection with CPB probed with the indicated antibodies. Tubulin served as loading control. (**D**) Representative micrographs of THP-1∆CD31gR.1 and THP-1∆gR.NT cells during a selection process using two consecutive incubations with CPB (1 μg/mL, 24 h, 37 °C) (200× magnification). (**E**) TIDE analysis of THP-1∆CD31gR.1 before and after selection with CPB. As control, cells transduced with gR.NT were used. (**F**) Western blots of THP-1∆gR.NT and THP-1∆CD31gR.1 before and after selection with CPB probed with the indicated antibodies. Tubulin served as loading control.

**Figure 5 toxins-13-00893-f005:**
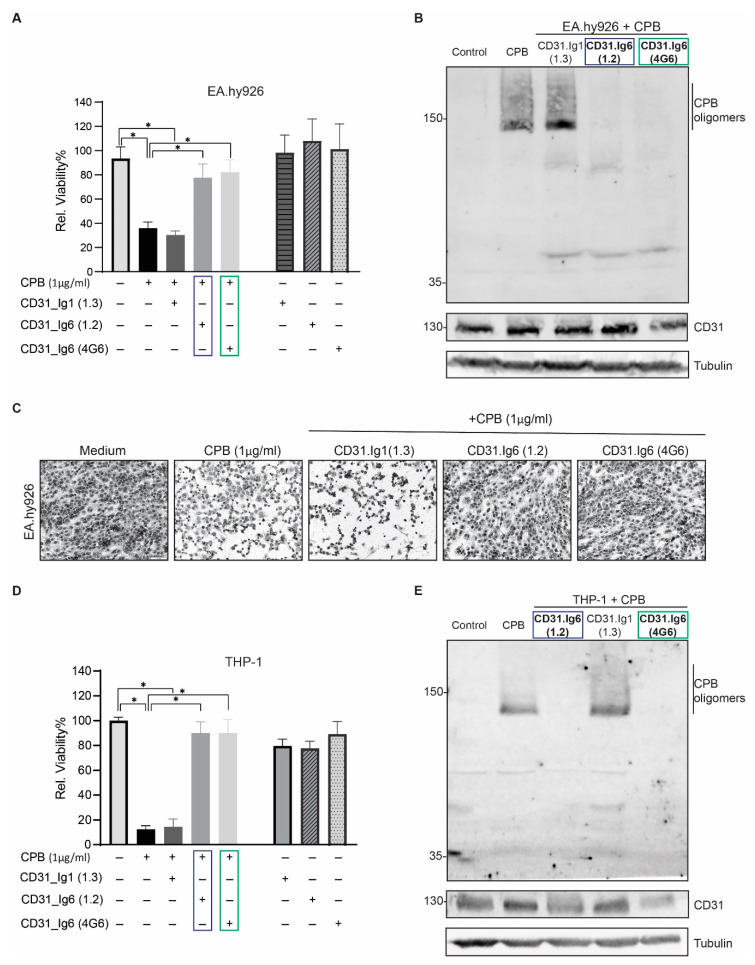
Blocking the CD31 Ig6 domain protects cells from CPB-mediated cytotoxicity. (**A**) Viability of EA.hy926 cells pre-incubated with anti-CD31 Ig1 and Ig6 domain monoclonal antibodies for 1 h/37 °C followed by addition of 1 μg/mL CPB for 24 h, 37 °C. Data are represented as means (*n* = 10) ± SD. Multiple comparison two-way ANOVA, Sidak’s multiple comparison test, values with *p* < 0.0001 are indicated by an asterisk. (**B**) Western blots of EA.hy926 cells pre-incubated with mAbs as in A, exposed to 8 μg/mL CPB for 20 min at 37 °C and probed with anti-CPB and anti-CD31 antibodies. Tubulin served as loading control. (**C**) Representative micrographs of EA.hy926 cells pre-incubated with anti-CD31 Ig1 and Ig6 domain monoclonal antibodies for 1 h/37 °C followed by addition of 1 μg/mL CPB for 24 h/37 °C. Scale bar, 100 μm. (**D**) Viability of THP-1 cells pre-incubated with anti-CD31 Ig1 and Ig6 domain monoclonal antibodies for 1 h/37 °C followed by addition of 1 μg/mL CPB for 24 h, 37 °C. Data are represented as means (*n* = 10) ± SD. Multiple comparison two-way ANOVA, Sidak’s multiple comparison test, values with *p* < 0.0001 are indicated by an asterisk. (**E**) Western blots of THP-1 cells pre-incubated with mAbs as in D, exposed to 8 μg/mL CPB for 60 min at 37 °C and probed with anti-CPB and anti-CD31 antibodies. Tubulin served as loading control. CD31 Ig6 domain mAbs are highlighted in a blue box (1.2) and a green box (4G6).

## Data Availability

Not applicable.

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
