# Peer review of "Platelet Endothelial Cell Adhesion Molecule 1 (CD31) Is Essential for Clostridium perfringens Beta-Toxin Mediated Cytotoxicity in Human Endothelial and Monocytic Cells"

_toxins, 2021, doi:10.3390/toxins13120893_

Round 1

Reviewer 1 Report

The paper appears to be an extension of a previous publication by the authors, this time applied to human endothelial cells. The research does not seem particularly novel, but the data appears solid.

I have several questions however:

Dots is Fig. 1A appear to be wrong. First point in THP-1 is lower than zero conc. Data appears not been collected at same concs for all cells. Is that correct? Why?

Fig. 1B data (CD31 expression) does not correlate well with viability in Fig. 1A. eg, 926 is more susceptible than HMEC1, but it expresses less CD31, whereas HMEC-1 is less susceptible than  THP-1, which appears to have much less CD31. Please provide explanation.  

In Fig1 A, viabilities of all cells are essentially 0 at 1 ug/mL, but in Fig. 2D, gr.NT versions have 30, 50% and even 70% viability (U937) at this conentration. How is this explained?

Panels above in Fig. 2E unclear. What is the band at 150 and the one above 35? What is the panel labeled ‘CPB’? Details should be included.

Line 204- what is the oligomeric state of CPB in solution? If monomeric, surely that form binds the receptor.

Minor grammar and punctuation:

Line 33 – produced, better than ‘synthesized’

\line 47 – most likely accounts for the targeting-> predicts the targeting

Line 55 – remove the commas before ‘and’ throughout the manuscript

Fig. 1A – x axis legend should be ng/mL (not clear)

X axis scale is odd. Please use log 10 scale: 0,1,10,100,1000 etc. Same in 3B and 3D

Line 58 – Expression levels… sentence unclear, please rewrite.

Line 72 – why only in HMEC-1 cells?

Reference to Fig. 2D is missing

IN Fig. 2D, viabilities are > 100%

Line 90 – separate number and units

Line 278 - ..? 30’000?

Some of the references, eg 45 to 47, Journal names are not abbreviated.

Reviewer 2 Report

Please confirm the attached file.
